# Trustworthy diagnosis of Electrocardiography signals based on out-of-distribution detection

Bowen Yu [1,2], Yuhong Liu[3], Xin Wu[4], Jing Ren[4], Zhibin Zhao [1]*

**1** Global Health Research Center, Duke Kunshan University, Jiangsu, China, **2** School of Mechanical Engineering, Xi'an Jiaotong University, Xi'an, China, **3** School of Big Health and Intelligent Engineering, Chengdu Medical College, Chengdu, China, **4** The First People's Hospital of Kunshan, Kunshan, Jiangsu, China

* zhaozhibin@xjtu.edu.cn

## Abstract

Cardiovascular disease is one of the most dangerous conditions, posing a significant threat to daily health. Electrocardiography (ECG) is crucial for heart health monitoring. It plays a pivotal role in early heart disease detection, heart function assessment, and guiding treatments. Thus, refining ECG diagnostic methods is vital for timely and accurate heart disease diagnosis. Recently, deep learning has significantly advanced in ECG signal classification and recognition. However, these methods struggle with new or Out-of-Distribution (OOD) heart diseases. The deep learning model performs well on existing heart diseases but falters on unknown types, which leads to less reliable diagnoses. To address this challenge, we propose a novel trustworthy diagnosis method for ECG signals based on OOD detection. The proposed model integrates Convolutional Neural Networks (CNN) and Attention mechanisms to enhance feature extraction. Meanwhile, Energy and ReAct techniques are used to recognize OOD heart diseases and its generalization capacity for trustworthy diagnosis. Empirical validation using both the MIT-BIH Arrhythmia Database and the INCART 12-lead Arrhythmia Database demonstrated our method's high sensitivity and specificity in diagnosing both known and out-of-distribution (OOD) heart diseases, thus verifying the model's diagnostic trustworthiness. The results not only validate the effectiveness of our approach but also highlight its potential application value in cardiac health diagnostics.

## 1 Introduction

Electrocardiography (ECG) plays a pivotal role in monitoring heart health, offering non-invasive insights into the electrophysiological activities of the heart. Subtle changes in these signals can be early indicators of heart disease, rendering ECG an indispensable tool for early disease detection, cardiac function evaluation, and guiding clinical treatments. With the global prevalence of cardiovascular diseases on the rise, particularly among aging populations, the need for efficient and accurate ECG diagnostics is more urgent than ever [1].

Traditionally, ECG diagnostics have relied heavily on manual feature extraction by clinicians, a process that is time-consuming, prone to subjective bias, and often limited in

**Data availability statement:** All relevant data are available from the MIT-BIH Arrhythmia Database. The dataset can be accessed at https://physionet.org/content/mitdb/1.0.0/ with accession number(s) 100, 101, 102 after acceptance.

**Funding:** This work was funded by The research results of this article (or publication) are sponsored by E Fund Global Health Lab, funding awarded to Z.Z.

**Competing interests:** The authors have declared that no competing interests exist.

accuracy. Traditionally, ECG diagnostics have relied on manual feature extraction by clinicians—a process that is not only time-consuming but also prone to subjective bias and limited accuracy. Recent advances in deep learning, however, have revolutionized ECG analysis by enabling models to autonomously learn complex patterns directly from raw signals, thus overcoming many limitations of traditional methods and significantly improving both accuracy and scalability [2,3]. For instance, Acharya et al. [4] pioneered the application of convolutional neural networks (CNNs) for arrhythmia classification, demonstrating that CNNs can surpass traditional methods in terms of both accuracy and processing speed. Building on this foundation, Kiranyaz et al. [5] further developed a 1D CNN architecture that processes raw ECG signals, achieving state-of-the-art performance in detecting various arrhythmia types. Additionally, recent studies have also explored the effectiveness of unsupervised pre-trained filter learning approaches in improving the efficiency of CNNs by reducing the reliance on large labeled datasets, thus enhancing model performance [6].

Beyond CNNs, other architectures, such as recurrent neural networks (RNNs), particularly Long Short-Term Memory (LSTM) networks, have shown promise in ECG analysis. Yildirim et al. [7] employed an LSTM network for ECG classification, demonstrating its ability to capture temporal dependencies in sequential data. In a similar vein, Sowmya and Jose [8] enhanced this approach by combining LSTM with CNN to create a hybrid model that leverages both spatial and temporal features, resulting in improved arrhythmia detection performance. Gao et al. [9] introduced an LSTM model to address class imbalance in ECG datasets, showing improved performance in detecting rare arrhythmias. Additionally, Warrick and Homsi [10] combined CNNs and LSTMs for arrhythmia detection, with the CNN component extracting temporal features and the LSTM capturing long-range dependencies. Hybrid deep CNN models, as proposed by other researchers, have also demonstrated strong performance in detecting abnormal arrhythmias, highlighting the importance of combining CNN with additional architectures for enhanced feature extraction [11].

Recently, transformer-based models have gained traction in the field of ECG analysis due to their superior ability to capture long-range dependencies, surpassing the capabilities of traditional RNNs. Park et al. [12] proposed a self-attention-based LSTM-FCN model for ECG signal classification, achieving competitive performance while also providing an uncertainty assessment. Similarly, Akan et al. [13] introduced ECGformer, a transformer architecture that leverages self-attention mechanisms to capture both local and global patterns in ECG signals, demonstrating superior performance in arrhythmia detection. Sun et al. [14] explored the application of transformer models in EEG signal classification, highlighting the potential of models like BERT to capture long-range dependencies in time-series biological signals and suggesting similar applications in ECG analysis.

In addition to architectural advances, various optimization techniques have been explored to boost model performance. Rajpurkar et al. [3] developed an optimized CNN model for ECG classification, achieving expert-level results. Zhu et al. [15] enhanced this with a Squeeze-and-Excitation (SE) residual network, improving the model's focus on relevant features in ECG signals.

Evolutionary methods like ModPSO-CNN [16] and CSFL [17] have shown potential in optimizing CNN parameters in other fields, though their application in ECG diagnostics is less explored. Unsupervised learning and evolutionary algorithms [18–20] have been effective in visual classification, offering promise for ECG signal analysis by enabling adaptive networks that generalize better to unseen, out-of-distribution (OOD) data. Techniques such as particle swarm optimization (PSO) have been used to fine-tune CNN architectures in ECG classification [21,22]. These methods dynamically adjust hyperparameters, improving robustness against noisy or imbalanced data, and leading to more accurate and reliable diagnostic outcomes.

Despite the remarkable success of these deep learning models in classifying known heart diseases, their effectiveness is largely confined to in-distribution (ID) data—conditions represented in the training set. When confronted with OOD heart diseases, which were not encountered during training, these models tend to produce overconfident yet inaccurate predictions, posing a significant challenge in clinical settings, as illustrated in Fig 1. This is because deep learning models, particularly CNNs and RNNs, are designed to excel at learning patterns present in the training data, meaning they can struggle when encountering data that deviates from this distribution. Hendrycks and Gimpel [23] highlighted this issue in their foundational work on OOD detection, showing that neural networks tend to make overconfident predictions when presented with OOD data, which can lead to dangerous misdiagnoses in medical applications.

The difficulty in handling OOD samples stems from several technical challenges. First, domain shifts between the training and testing data—such as variations in patient demographics or different ECG recording environments—can lead to performance degradation. Heart disease manifests differently across populations, and ECG signals may vary based on factors such as age, gender, and pre-existing conditions. These domain shifts can cause a model trained on one cohort to perform poorly on another [24,25]. Specifically, in ECG datasets, small variations in signal noise, recording conditions, or patient demographics can significantly degrade model performance, leading to incorrect diagnoses for both known and unknown heart conditions.

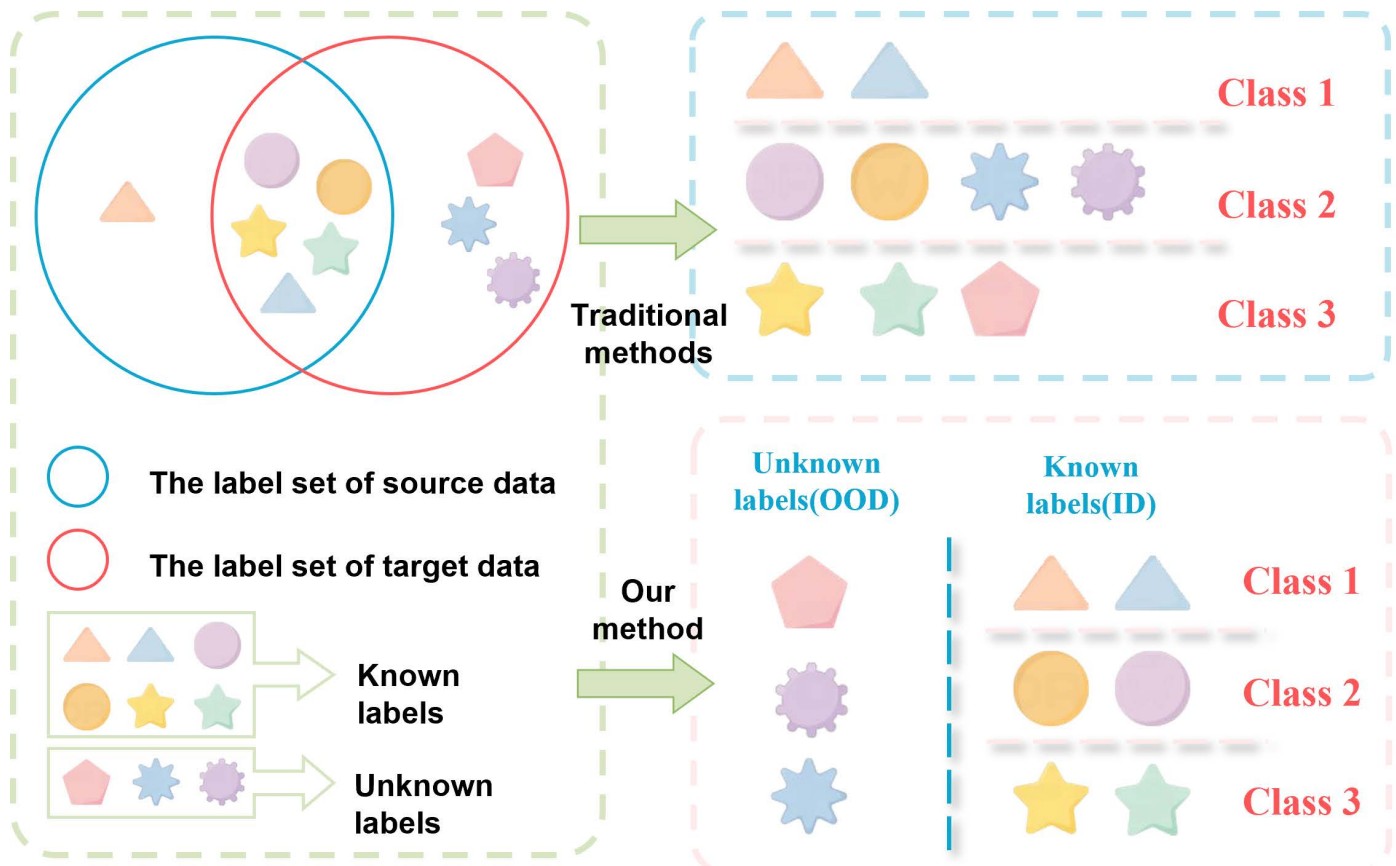

**Fig 1. Illustrations of differences between traditional and our method.**

Class imbalance is another challenge in ECG datasets. Rare arrhythmias are often under-represented, making it difficult for models to learn robust features for these conditions. Choi et al. [26] addressed this issue by employing data augmentation techniques, such as oversampling minority classes and generating synthetic ECG signals using generative adversarial networks (GANs). Esteban et al. [27] also introduced a synthetic data generation approach using variational autoencoders (VAEs) to tackle class imbalance, though their models still struggled with OOD cases. These challenges highlight the importance of improving models not only in terms of classification accuracy for known diseases but also in their ability to reliably detect and flag unknown conditions.

Recent studies have explored anomaly detection techniques as a potential solution to address the OOD problem. Shin et al. [28] developed an improved AnoGAN model for arrhythmia detection, while Hossain et al. [29] introduced ECG-Adv-GAN, a GAN-based approach for generating realistic ECG signals and detecting abnormalities. Qin et al. [30] proposed a temporal generative adversarial network for time-series anomaly detection. However, these approaches primarily focus on identifying anomalies without providing accurate classification of known diseases, limiting their practical utility in clinical diagnostics where both tasks are essential.

To address these limitations, we propose a novel method that integrates Energy-based OOD detection and ReAct, a technique designed to mitigate overconfidence in neural networks. Liu et al. [31] introduced Energy-based OOD detection, which calculates the energy of neural network outputs to distinguish between ID and OOD samples. By computing the energy score for each sample, our model can better discern whether a given input belongs to the training distribution, reducing the likelihood of overconfident predictions on OOD samples. ReAct, introduced by Sun et al. [32], is a complementary technique that modifies neural activations to dampen overconfident predictions, particularly in OOD scenarios. By clipping the activations of the penultimate layer, ReAct reduces the model's tendency to make overly confident predictions on unfamiliar inputs.

Our approach combines these techniques with a CNN-Attention mechanism to enhance feature extraction and improve the model's sensitivity to OOD samples. The CNN-Attention architecture leverages the spatial feature extraction capabilities of CNNs and the attention mechanism's ability to focus on the most relevant parts of the signal, ensuring both accurate classification of known conditions and reliable detection of unknown ones.

By integrating these advanced techniques, our method not only improves classification performance on ID data but also ensures robust detection of OOD heart diseases, addressing a significant gap in the current state of ECG analysis.

Our method contributes to the field in several key ways:

Dual-Mode Diagnosis: We develop a system capable of both accurately classifying known heart diseases and effectively detecting unknown ones, addressing a significant gap in current ECG diagnostic technologies.

Enhanced OOD Detection: By integrating Energy and ReAct techniques, our method improves over traditional Softmax classifiers, which tend to produce overly confident predictions on OOD samples. This ensures more trustworthy diagnostic outcomes, particularly in clinical scenarios involving unknown heart diseases.

Real-World Validation: We validated our method on two widely used ECG datasets—the MIT-BIH Arrhythmia Database and the INCART 12-lead Arrhythmia Database. This real-world evaluation highlights the robustness and practical applicability of our approach in providing comprehensive and reliable ECG-based diagnostics.

The rest of this paper is organized as follows: Section II details the methods and materials, emphasizing the importance of trustworthy diagnosis and OOD detection in ECG analysis

and explains how Energy, ReAct, CNN, and Attention mechanisms are integrated to form a reliable diagnostic framework. Section III presents experimental results, showcasing the effectiveness of our approach in delivering trustworthy diagnostic solutions. Finally, Section V summarizes the research findings and outlines potential directions for future work in ECG-based diagnostics.

## 2. Methods and datasets

### 2.1 Overview of the proposed method

In OOD Detection, the label information of samples in the training set is known while the test set involves samples with unknown labels not included in the training set. In this context, this study is to identify newly emerging abnormal heart diseases in the test set. The training set $\mathcal{D}_{\text{train}} = \{(x_i, y_i)_{i=1}^{n_s}\}$ consists of $n_s$ labeled samples, encompassing $\mathcal{C}_{\text{train}}$ known heart disease categories. Meanwhile, the test set $\mathcal{D}_{\text{test}} = \{(x_j)_{j=1}^{n_t}\}$ comprises $n_t$ labeled samples covering $\mathcal{C}_{\text{test}}$ types of heart diseases. Notably, under the assumption that $\mathcal{C}_{\text{test}}$ contains heart disease categories not included in $\mathcal{C}_{\text{train}}$, i.e., $\mathcal{C}_{\text{train}} \leq \mathcal{C}_{\text{test}}$, $\mathcal{C} = \mathcal{C}_{\text{test}} / \mathcal{C}_{\text{train}}$ represents the unknown heart disease categories. Therefore, the proposed OOD detection approach focuses on effectively identifying both known and unknown heart diseases in the test set.

In this study, we propose a novel, end-to-end deep learning-based system for trustworthy arrhythmia diagnosis using ECG signals. As demonstrated in Fig 2, our model is designed to accurately classify known heart diseases while also detecting OOD conditions, representing heart diseases that were not part of the training set. This is crucial for reliable clinical

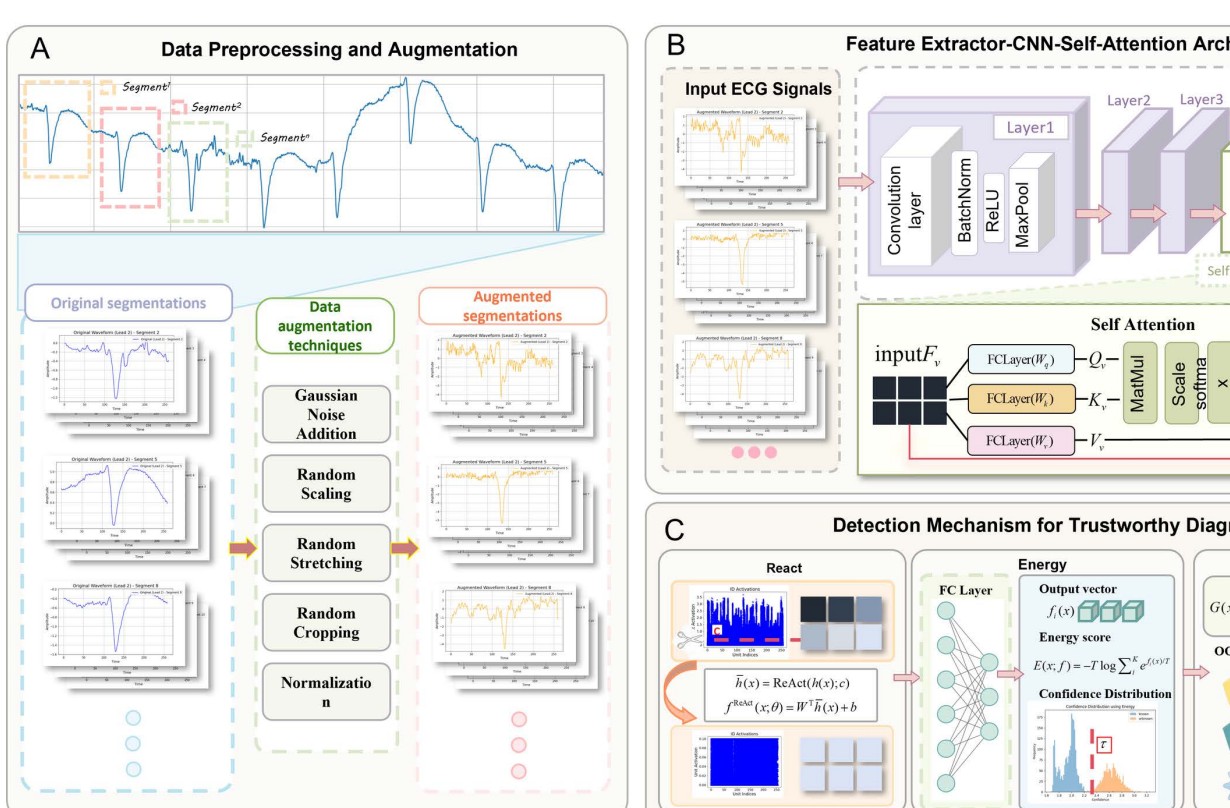

**Fig 2. Framework of the proposed method.**

applications, as it ensures that the model does not make overconfident predictions on novel or unseen conditions.

The core components of our proposed method include:

**Data Preprocessing and Augmentation:** Raw ECG signals are first segmented into fixed-length windows, and various data augmentation techniques are applied to enhance model robustness.

**CNN-Self Attention Architecture:** A hybrid architecture combining CNNs for feature extraction and a Self-Attention mechanism to focus on the most relevant parts of the ECG signal. This architecture allows the model to effectively capture both local features and global relationships within the ECG data.

**Detection Mechanism for Trustworthy Diagnosis:** This component integrates the Energy-based scoring mechanism for OOD detection to assess whether an input belongs to an out-of-distribution class. Additionally, the ReAct mechanism is applied to mitigate overconfidence in OOD predictions by truncating activations in the penultimate layer, ensuring safer and more reliable diagnostic outcomes.

After integrating these components, the model is trained and evaluated on two widely used ECG datasets—the MIT-BIH Arrhythmia Database and the INCART 12-lead Arrhythmia Database—to demonstrate its effectiveness in both ID classification and OOD detection.

## 2.2  CNN-SelfAttention architecture

The core of our model is the CNN-SelfAttention architecture, designed to extract both local features (e.g., P, Q, R, S, and T peaks) and global features (e.g., the relationships between successive heartbeats) from ECG signals. This hybrid architecture combines the spatial feature extraction power of Convolutional Neural Networks (CNNs) with the Self-Attention mechanism's ability to capture long-range dependencies in the data.

**CNN Component.** The CNN component is responsible for extracting hierarchical features from the raw ECG signal. It consists of three convolutional layers, each followed by batch normalization, ReLU activation, and max-pooling operations. These layers progressively extract low-level and high-level features from the ECG signal while reducing the dimensionality of the input. The specifics of these layers are shown in Table 1:

After passing through these convolutional layers, the ECG signal is transformed into a compact, high-level feature representation. These features are then flattened and fed into a fully connected layer, reducing the dimensionality to 64, which serves as the input to the subsequent Self-Attention mechanism.

**Self-attention mechanism.**  Once the features are extracted and reduced in dimensionality, they are passed through a Self-Attention mechanism, which plays a crucial role in capturing long-range dependencies and relationships between different parts of the ECG signal. This is essential for identifying complex arrhythmias that may span multiple heartbeats. The Self-Attention mechanism consists of the following steps:

**Table 1.  Structure of the CNN component for feature extraction.**

| Layer | Operation | Filters/Units | Kernel Size | Stride | Padding | Activation | Pooling | Dropout |
|---|---|---|---|---|---|---|---|---|
| Layer1 | 1D Convolutional Layer | 16 | 5 | 1 | 2 | ReLU | Max Pooling (2, 2) | 0.2 |
|  | Batch Normalization | N/A | N/A | N/A | N/A | N/A | N/A | N/A |
| Layer2 | 1D Convolutional Layer | 32 | 3 | 1 | 1 | ReLU | Max Pooling (2, 2) | 0.2 |
|  | Batch Normalization | N/A | N/A | N/A | N/A | N/A | N/A | N/A |
| Layer3 | 1D Convolutional Layer | 64 | 3 | 1 | 1 | ReLU | Adaptive Max Pooling (1) | N/A |
|  | Batch Normalization | N/A | N/A | N/A | N/A | N/A | N/A | N/A |

**1. Query, Key, Value Projection:** The input feature matrix $X \in \mathbb{R}^{n \times d_{in}}$, where $n$ is the number of input tokens (in this case, the sequence length of the ECG signal) and $d_{in}$ is the input feature dimension (here, 64), is transformed into three different linear projections: Query $Q$, Key $K$ and Value $V$.

$$Q = XW_Q, K = XW_K, V = XW_V \tag{1}$$

Where $W_Q \in \mathbb{R}^{d_{in} \times d_k}$, $W_K \in \mathbb{R}^{d_{in} \times d_k}$ and $W_V \in \mathbb{R}^{d_{in} \times d_k}$ are the learnable weight matrices, and $Q \in \mathbb{R}^{n \times d_k}$, $K \in \mathbb{R}^{n \times d_k}$ and $V \in \mathbb{R}^{n \times d_k}$ are the resulting query, key, and value matrices.

**2. Calculating the Attention Score Matrix:** The attention score matrix $A \in \mathbb{R}^{n \times n}$ is computed as the dot product between the query and the transpose of the key matrices, followed by scaling based on the dimension of the key:

$$A_{ij} = \frac{Q_i K_j^{\mathrm{T}}}{\sqrt{d_k}} \tag{2}$$

Where $d_k$ is the dimension of the key and the scaling factor $\sqrt{d_k}$ prevents the dot product values from becoming too large.

**3. Softmax Function:** The softmax function is applied to the attention score matrix $A$ to obtain the attention matrix $\alpha \in \mathbb{R}^{n \times n}$, which converts the scores into a probability distribution:

$$\alpha_{ij} = \mathrm{softmax}(A_{ij}) = \frac{\exp(A_{ij})}{\sum_k \exp(A_{ij})} \tag{3}$$

This ensures that the attention weights for each query sum to 1, representing the relative importance of each key with respect to a given query.

**4. Weighted Value Matrix:** The weighted value matrix $Z \in \mathbb{R}^{n \times d_k}$ is computed by multiplying the attention matrix $\alpha$ with the value matrix $V$:

$$Z_i = \sum_j a_{ij} V_j \tag{4}$$

**5. Residual Connection:** Finally, a residual connection is applied by summing the input features $X$ with the attention output $Z$:

$$O = Z + X \tag{5}$$

This residual connection helps preserve the original input features while incorporating the information captured by the Self-Attention mechanism. It ensures that the model retains local information while also capturing global dependencies, which is crucial for accurate arrhythmia detection.

**Final classification layer.** After the Self-Attention mechanism, the output features are passed through the following fully connected layers for the final classification, of which the structure is detailed in Table 2.

The final prediction is a classification of the input ECG signal into one of the predefined arrhythmia classes. This architecture is designed to effectively capture both local and global features of the ECG signal, enabling accurate arrhythmia detection and classification.

## 2.3 Detection mechanism for trustworthy diagnosis

To achieve trustworthy diagnoses, we incorporate two key mechanisms: Energy-based OOD detection and the ReAct mechanism. Together, these approaches mitigate the problem of

**Table 2. Structure of the final classification layers.**

| Layer | Operation | Description |
|---|---|---|
| Dropout Layer | Dropout | Dropout rate of 0.2 |
| Fully Connected Layer | Linear Layer | Reduces the dimensionality from 64 to 32. |
| ReLU Activation | Non-linear Activation | Introduces non-linearity. |
| Output Layer | Linear Layer | Maps the features to the number of output classes |

overconfident predictions on OOD samples, enhancing the model's ability to distinguish between known and unknown heart conditions, and ultimately ensuring safer, more reliable diagnostic results.

**2.3.1 Energy score for OOD detection.** The Energy Score is employed to quantify the likelihood that a given input belongs to an OOD class, offering a more reliable alternative to the traditional confidence scores generated by the Softmax function. The Energy-based method is less prone to overconfidence, making it a better fit for OOD detection and contributing to the trustworthiness of the overall diagnosis. The core of the energy model is the energy function $E(x)$, mapping each point $x$ in the input space to a non-probabilistic scalar, i.e., the energy. Through the Gibbs distribution, a set of energy values can be transformed into a probability density $p(x)$:

$$p(x) = \frac{\exp(-E(x)/T)}{Z(T)} \tag{6}$$

where $Z(T)$ is called the partition function, marginalizing over all possible states, and $T$ is the temperature parameter. The energy for a given data point is expressed as the negative log of the partition function:

$$E(x) = -\log Z(T) \tag{7}$$

In this formulation, the discriminative neural network model maps an input $x$ to a set of real-valued logits. The Energy score theoretically aligns with the probability density of in-distribution data, offering a significant advantage over traditional Softmax-based confidence scores. However, the effectiveness of the Energy-based method depends heavily on the characteristics of the data and the model. In cases where the energy gap between in-distribution and OOD data is insufficient for accurate differentiation, further refinement of the Energy method is necessary.

**2.3.2 ReAct mechanism.** The ReAct mechanism is designed to reduce overconfidence in predictions, particularly when handling OOD samples. It works by truncating the activations in the penultimate layer of the neural network, capping the activation values at a specified threshold $c > 0$. This operation can be applied to a pre-trained model without any modifications to the training process, making it a flexible approach for improving model robustness.

Fig 3 shows the distribution of activations in the penultimate layer of CNN-Attention trained on the MIT-BIH dataset. The ReAct threshold was determined using a grid search approach. Specifically, candidate thresholds were selected based on different percentiles of the activation values from the ID dataset in the penultimate layer, ranging from the 5th percentile to the 100th percentile in increments of 5%. For each threshold, we evaluated the model's performance on a test set containing both ID and OOD samples. The threshold that minimized Detection Error (DE) while maximizing AUROC and AUPR was selected as the optimal value.

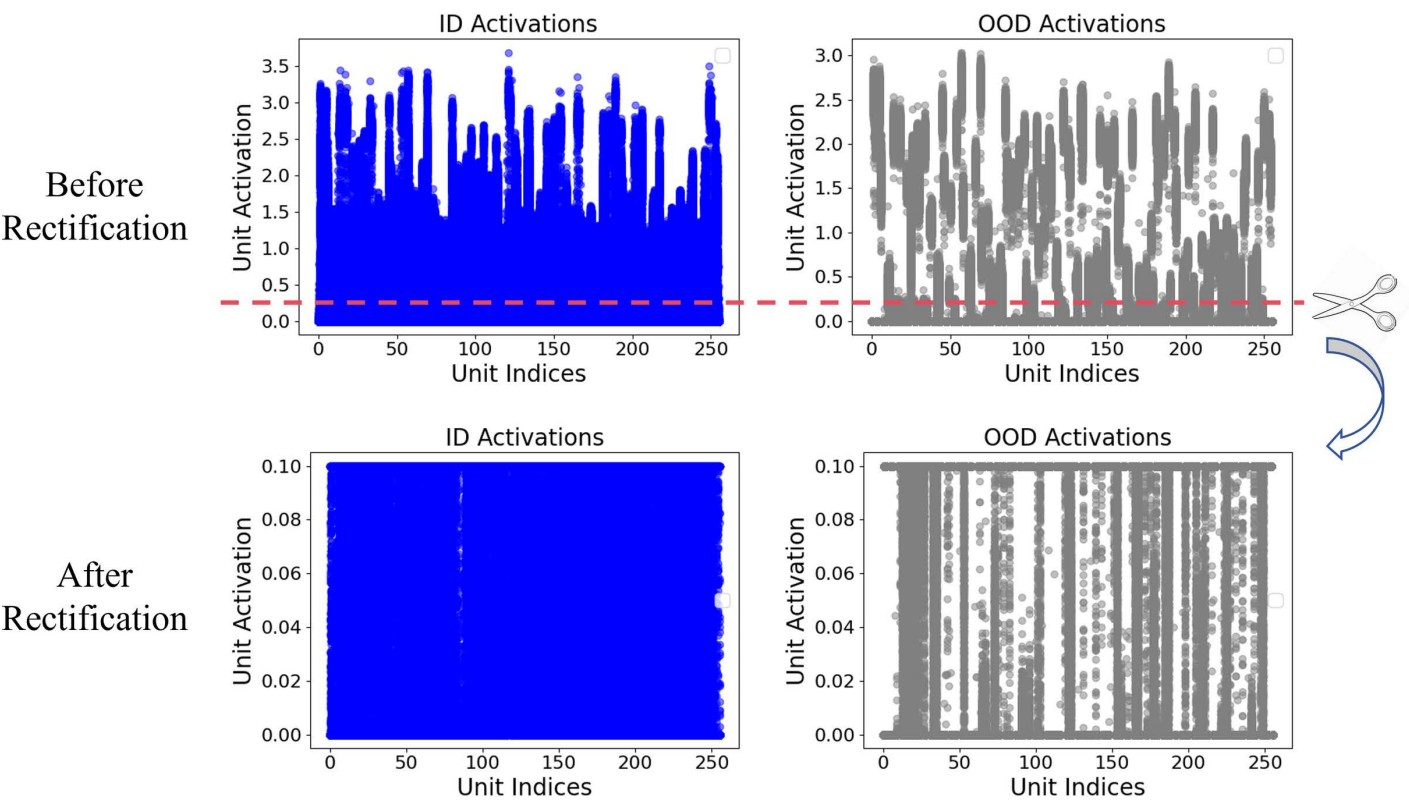

**Fig 3. The distribution of activations in the penultimate layer for ID and OOD data.**

For a given input $x$ and its corresponding feature vector from the penultimate layer $h(x)$, the ReAct operation is defined as:

$$\overline{h}(x) = \mathrm{ReAct}(h(x); c) \tag{8}$$

where $\mathrm{ReAct}(x; c) = \min(x, c)$ and is applied element-wise to the feature vector $h(x)$ and $c$ is the upper limit of activation values. This rectification process helps prevent the network from making overly confident predictions for OOD samples, while maintaining high classification accuracy for in-distribution data. After applying ReAct, the model's output is:

$$f^{\mathrm{ReAct}}(x; \theta) = W^{\mathrm{T}}\overline{h}(x) + b \tag{9}$$

where $W$ and $b$ are the weight matrix and bias vector, respectively, and $\theta$ represents the network parameters. By limiting the magnitude of the activations, ReAct effectively reduces the tendency of the model to make overconfident predictions on unfamiliar inputs, thereby improving the trustworthiness of the diagnostic system.

**2.3.3 Decision process for comprehensive diagnosis.** To ensure comprehensive and trustworthy diagnosis, we set an energy threshold for classifying whether a sample is OOD. This threshold is determined by minimizing the detection error (DE), which accounts for both false positives and false negatives. Samples with energy scores below this threshold are flagged as OOD, signaling potential unrecognized heart disease categories, while samples with energy scores above the threshold are classified based on known heart diseases using the CNN-Attention model's predictive capabilities.

By combining the Energy-based OOD detection mechanism with the ReAct technique, our method provides a holistic and reliable diagnostic solution, accurately classifying known heart conditions while effectively identifying unknown ones. This dual approach ensures that both in-distribution and out-of-distribution samples are handled appropriately, leading to safer and more trustworthy diagnostic recommendations.

$$G(x; \tau, f) = \begin{cases} 0, & \text{if } -E(x; f) \leq \tau, \\ 1, & \text{if } -E(x; f) > \tau. \end{cases} \quad (10)$$

## 2.4 Dataset introduction

To evaluate the effectiveness and generalization capability of our proposed method, we utilized two well-established ECG datasets: the MIT-BIH Arrhythmia Database and the INCART 12-lead Arrhythmia Database. These datasets provide a diverse range of arrhythmia patterns and serve as common benchmarks in ECG analysis. Below, we detail the characteristics of each dataset, including the specific arrhythmia categories selected and the approach to data splitting.

**2.4.1 MIT-BIH arrhythmia database.** The MIT-BIH Arrhythmia Database is one of the most commonly used datasets for ECG analysis and arrhythmia classification, containing 48 half-hour recordings from 47 subjects. Each recording was sampled at 360 Hz and contains two ECG leads. For this study, we only used Lead II for consistency and to facilitate model training across different datasets.

The dataset provides annotations for 15 different types of arrhythmias, but for our analysis, we selected the following five major arrhythmia categories:

N: Normal beat

A: Atrial Premature Beat

R: Right Bundle Branch Block

V: Premature Ventricular Contraction

L: Left Bundle Branch Block

These categories cover both common and clinically significant arrhythmias, allowing us to evaluate the model's performance across a representative range of heart conditions.

**2.4.2 INCART 12-lead Arrhythmia database.** The INCART 12-lead Arrhythmia Database contains 75 annotated ECG recordings, sampled at 257 Hz, collected in clinical settings. Each recording includes 12 synchronously recorded leads. As with the MIT-BIH dataset, we used Lead II for consistency and ease of integration between datasets. The INCART dataset provides a more diverse set of arrhythmias, which enhances the model's robustness and generalization capabilities.

For this study, we selected the following four arrhythmia categories from the INCART dataset:

N: Normal beat

A: Atrial Premature Beat

R: Right Bundle Branch Block

V: Premature Ventricular Contraction

These categories were chosen to align with the MIT-BIH dataset, allowing for a consistent evaluation of the model's performance across different datasets.

**2.4.3 Dataset splitting.** To prevent data leakage and artificially inflated performance, both the MIT-BIH and INCART datasets were split based on patient IDs rather than individual heartbeats. This ensures that heartbeats from the same patient do not appear in both the training and test sets, which would otherwise lead to overfitting and inflated accuracy due to the similarity of heartbeats from the same individual.

MIT-BIH Dataset

For the MIT-BIH dataset, we specifically selected recordings from 28 patients for training, 10 patients for validation, and 10 patients for testing. The patient-specific file names used in each split and the number of different arrhythmia categories are shown in Table 3.

The patient-based splitting strategy for the INCART dataset is detailed in Table 4, ensuring that recordings from the same patient are confined to a single subset (training, validation, or test). The specific file names for each split and the number of different arrhythmia categories are as follows:

To further evaluate the model's ability to handle OOD samples, we designed specific tasks that involved categorizing known and unknown arrhythmia types. The tasks are described in Table 5.

**2.4.4 Data preprocessing and augmentation.** For both datasets, we employed a window-based approach to segment the ECG signals into fixed-length windows of 260 milliseconds, which is approximately the duration of one heartbeat. Unlike traditional methods that rely on R-peak detection for segmentation, we directly segmented the raw ECG signals into windows of fixed length, following the end-to-end deep learning approach described in Park et al. (2023). This method allows the model to learn both temporal and morphological features from the raw signal without relying on handcrafted preprocessing steps, such as R-peak detection.

**Table 3. Patient-based splitting strategy for the MIT-BIH dataset.**

| Task | Known/Unknown | N | A | R | V | L |
|---|---|---|---|---|---|---|
| Training set | '109', '219', '112', '102', '234', '101', '210', '105', '214', '104', '217', '115', '107', '200', '106', '124', '123', '222', '212', '203', '213', '114', '221', '207', '209', '223', '108', '121' | 41593 | 854 | 3440 | 3919 | 5948 |
| Validation set | '220', '118', '116', '215', '202', '232', '117', '111', '103', '119' | 14663 | 1615 | 2562 | 753 | 2123 |
| Test set | '113', '233', '208', '230', '231', '201', '205', '228', '100', '122' | 18765 | 77 | 1254 | 2457 | 0 |

**Table 4. Patient-based splitting strategy for the INCART dataset.**

| Task | Known/Unknown | N | A | R | V |
|---|---|---|---|---|---|
| Training set | 'I16', 'I18', 'I19', 'I20', 'I14', 'I15', 'I13', 'I12', 'I21', 'I27', 'I28', 'I29', 'I30', 'I33', 'I04', 'I36', 'I37', 'I38', 'I39', 'I40', 'I49', 'I50', 'I51', 'I52', 'I53', 'I41', 'I42', 'I43', 'I44', 'I45', 'I56', 'I57', 'I58', 'I59', 'I60' | 68212 | 940 | 1520 | 11923 |
| Validation set | 'I46', 'I47', 'I48', 'I54', 'I55', 'I17', 'I23', 'I24', 'I25', 'I26', 'I31', 'I32', 'I34', 'I35', 'I01', 'I02', 'I03', 'I05', 'I11', 'I70' | 40619 | 712 | 8 | 3645 |
| Test set | I61', 'I62', 'I63', 'I64', 'I65', 'I66', 'I67', 'I68', 'I69', 'I22', 'I71', 'I72', 'I73', 'I74', 'I75', 'I06', 'I07', 'I08', 'I09', 'I10' | 41498 | 290 | 1645 | 4437 |

**Table 5. Tasks under different combinations of samples.**

| | Task | Known/Unknown | Train | Validation | Test ID | Test OOD |
|---|---|---|---|---|---|---|
| MIT-BIH | Task0 | - [A, R] | 51460 | 17539 | 21222 | 1331 |
| | Task1 | - [L, V] | 45887 | 18840 | 20096 | 2457 |
| INCART | Task3 | - [A] | 81655 | 44272 | 47580 | 290 |

In addition to segmentation, data augmentation techniques were applied to the training data to enhance the model's robustness and generalization capabilities. The following augmentation techniques were employed:

**Gaussian Noise Addition:** Random Gaussian noise with a standard deviation of $\sigma = 0.02$ is added to the signal to simulate real-world noise and improve robustness to noisy data.

Random Scaling: The amplitude of the ECG signal is scaled randomly with a factor drawn from a normal distribution with $\sigma = 0.02$, simulating variations in signal amplitude due to differences in electrode placement or patient physiology.

**Random Stretching:** The ECG signal is randomly stretched or compressed along the time axis with a factor drawn from a normal distribution with $\sigma = 0.01$, simulating variations in heart rate.

**Random Cropping:** A small portion of the ECG signal, with a length of 10 samples, is randomly removed to simulate missing data or sensor dropout.

**Normalization:** Normalizes the ECG signal to zero mean and unit variance, ensuring consistent signal scaling and improving model convergence during training.

These augmentation techniques allow the model to learn from a more diverse set of inputs, improving its ability to generalize to unseen data and handle real-world variability in ECG signals.

## 2.5 Experimental setup

To tackle the challenges posed by class imbalance in ECG data, particularly the rarity of abnormal heart disease samples, we adopted a loss function combining Cross-Entropy Loss and Focal Loss. This hybrid loss function allows the model to effectively learn from imbalanced data, improving classification accuracy for common arrhythmias while enhancing the model's sensitivity to Out-of-Distribution (OOD) samples.

**2.5.1 Loss functions.** The Cross-Entropy Loss (CE) was primarily used to optimize the model for standard classification tasks. It is defined as:

$$\mathcal{L}_{\mathrm{CE}} = -\sum_{i=1}^{N} y_i \log(p_i) \tag{11}$$

where $N$ is the number of samples, $y_i$ is the true label of sample $i$, and $p_i$ is the predicted probability for the correct class.

Since abnormal heart disease samples are relatively rare, we further introduce the Focal Loss [33] to mitigate the class imbalance issue and enhance the model's sensitivity to OOD samples, which is crucial for reliable OOD detection. The Focal Loss is defined as:

$$\mathcal{L}_{\mathrm{FL}} = -\sum_{i=1}^{N} \alpha (1 - p_i)^{\gamma} \log(p_i) \tag{12}$$

where $\alpha$ is a weighting factor that balances the importance of positive and negative samples, and $\gamma$ is a focusing parameter that reduces the contribution of easily classified samples and allows more focus on difficult cases.

By combining these two loss functions, we aim to optimize the model's performance across both frequent and rare arrhythmia classes. The final loss function used during training is a weighted sum of Cross-Entropy Loss and Focal Loss, expressed as:

$$\mathcal{L} = \mathcal{L}_{\mathrm{CE}} + \lambda \mathcal{L}_{\mathrm{FL}} \tag{13}$$

where $\lambda$ is a hyperparameter that balances the contributions of the two loss functions. This hybrid loss function enables the model to achieve high accuracy on ID samples while also improving its ability to detect and correctly classify OOD samples.

**2.5.2 Training configuration and hardware configuration.** The model was trained using the Adam optimizer with an initial learning rate of 0.001 and a weight decay of 1e-5 to prevent overfitting. The training process was carried out for a total of 30 epochs, with a batch size of 48. To further mitigate overfitting, dropout with a rate of 0.2 was applied to the fully connected layers.

We utilized a ReduceLROnPlateau scheduler to dynamically adjust the learning rate based on validation performance. The learning rate was reduced by a factor of 0.1 if the validation loss did not improve for 5 consecutive epochs. This ensured that the learning rate was appropriately reduced as the model approached convergence, preventing overfitting.

All experiments were conducted on a server equipped with NVIDIA Tesla V100 GPUs, providing the computational resources required for efficient training and evaluation.

## 3 Experimental results

To validate the effectiveness of our proposed method in delivering trustworthy diagnoses, we conducted extensive comparative experiments. Our approach integrates CNNs, Attention mechanisms, Energy-based OOD detection, and ReAct techniques. The method was rigorously evaluated using two widely recognized ECG datasets—MIT-BIH Arrhythmia Database and INCART 12-lead Arrhythmia Database. The experiments focused on both ID classification and OOD detection, ensuring a comprehensive assessment of the model's capabilities.

### 3.1 Out-of-distribution detection results

In the OOD detection experiments, we assessed the model's ability to identify heart diseases absent from the training data. Our method, which integrates Energy-based OOD detection with the ReAct technique, was compared against traditional methods like Softmax and state-of-the-art OOD detection methods such as ODIN [34]. The following key metrics were used to evaluate detection performance:

**Detection Error (DE):** The overall error rate, considering both false positives (in-distribution samples misclassified as OOD) and false negatives (OOD samples misclassified as in-distribution), calculated as $\text{DE} = \dfrac{1 - \text{TPR} + \text{FPR}}{2}$.

**False Positive Rate at 95% True Positive Rate (FPR95):** The proportion of in-distribution samples incorrectly classified as OOD when the true positive rate is fixed at 95%. This metric emphasizes the trade-off between sensitivity and specificity, which is crucial in clinical applications.

**Area Under the ROC Curve (AUROC):** This metric evaluates the model's ability to distinguish between in-distribution and OOD samples across various thresholds. A higher AUROC indicates better discriminative performance, calculated as $\text{AUROC} = \int_0^1 \text{TPR}(\text{FPR})\,\text{d}(\text{FPR})$

**Area Under the Precision-Recall Curve (AUPR):** This metric reflects the model's performance in identifying OOD samples, particularly in imbalanced settings where OOD samples are rare, calculated as. $\text{AUPR} = \int_0^1 \text{Precision}(\text{Recall})\,\text{d}(\text{Recall})$

**False Discovery Rate (FDR):** The proportion of samples predicted as OOD that are actually ID samples, calculated as $\text{FDR} = \dfrac{\text{FP}}{\text{FP} + \text{TP}}$.

Table 6 compares the performance of various methods on the MIT-BIH dataset for two tasks (Task 1 and Task 2). Our method, which combines Energy-based OOD detection with the ReAct technique, demonstrates significantly improved results across several metrics compared to methods like Softmax and ODIN.

In Task 1, our Energy+ReAct approach reduced the FPR95 to 5.71%, a substantial improvement compared to Softmax (99.92%) and ODIN (99.85%). Additionally, our method achieved an AUROC of 97.27%, outperforming both Softmax and ODIN, which recorded AUROCs of 70.39% and 71.18%, respectively. The AUPR (99.69%) and F1-Score (98.96%) were also the highest among the compared methods, demonstrating that Energy+ReAct provides a more reliable overall detection performance.

In Task 2, Energy+ReAct also outperformed other methods, achieving an AUROC of 84.94%, with a significantly lower DE and FDR compared to Softmax and ODIN. While the results in Task 2 were not as strong as in Task 1, the overall performance remains competitive. The reduction in FPR95 to 54.95% (from 79.20% for Softmax and 82.87% for ODIN) further highlights the value of our approachin OOD detection.

Similarly, as shown in Table 7, on the INCART dataset for Task 3, our Energy+ReAct method achieved the best overall performance across several metrics. Specifically, it attained an AUROC of 69.63%, which is significantly higher than the AUROC scores of Softmax (14.29%) and ODIN (43.68%). In addition to this, our method also achieved an AUPR of 99.75%, F1-Score of 99.70%, and Detection Error of 29.66%, outperforming Softmax and ODIN across all these metrics. These results clearly demonstrate that our model generalizes well across different datasets, even when the ECG signals are more complex or variable, as seen in the INCART dataset.

## 3.2 Ablation study results

To further investigate the contributions of different components in our architecture, we conducted an ablation study. Specifically, we evaluated the performance of the model by removing or modifying key components such as the energy-based scoring and ReAct mechanisms. Table 8 shows the results of the ablation study on the MIT-BIH and INCART datasets.

**Table 6. Detection performance using different methods with MIT-BIH.**

| Method | Task | FPR95 ↓ | AUROC ↑ | AUPR ↑ | DE ↓ | FDR ↓ |
|---|---|---|---|---|---|---|
| Softmax | Task1 | 99.92 | 70.39 | 97.72 | 24.66 | 5.90 |
| | Task2 | 79.20 | 71.84 | 95.18 | 33.35 | 10.74 |
| ODIN | Task1 | 99.85 | 71.18 | 97.81 | 22.68 | 5.90 |
| | Task2 | 82.87 | 77.34 | 96.24 | 24.15 | 10.83 |
| Energy⁺ React | Task1 | 5.71 | 97.27 | 99.69 | 3.89 | 0.45 |
| | Task2 | 54.95 | 84.94 | 97.66 | 21.23 | 7.93 |

**Table 7. Detection performance using different methods with INCART.**

| Method | Task | FPR95 ↓ | AUROC ↑ | AUPR ↑ | DE ↓ | FDR ↓ |
|---|---|---|---|---|---|---|
| Softmax | Task3 | 98.24 | 14.29 | 90.59 | 49.97 | 3.44 |
| ODIN | Task3 | 99.66 | 43.68 | 99.39 | 46.55 | 0.61 |
| Energy⁺ React | Task3 | 100 | 69.63 | 99.75 | 29.66 | 0.61 |

**Table 8. Ablation study results.**

| Method | Task | FPR95 ↓ | AUROC ↑ | AUPR ↑ | DE ↓ | FDR ↓ |
|---|---|---|---|---|---|---|
| Only Energy | Task1 | 5.71 | 95.98 | 99.62 | 3.89 | 0.53 |
| | Task2 | 57.31 | 54.30 | 97.51 | 21.16 | 8.05 |
| | Task3 | 98.62 | 64.36 | 99.67 | 35.85 | 0.61 |
| Only React | Task1 | 5.94 | 95.33 | 99.53 | 4.36 | 0.38 |
| | Task2 | 79.20 | 71.82 | 95.17 | 33.34 | 10.74 |
| | Task3 | 97.59 | 47.42 | 99.46 | 44.09 | 0.61 |
| Energy + React | Task1 | 5.71 | 97.27 | 99.69 | 3.89 | 0.45 |
| | Task2 | 54.95 | 84.94 | 97.66 | 21.23 | 7.93 |
| | Tsk3 | 100 | 69.63 | 99.75 | 29.66 | 0.61 |

Energy-based scoring enhances the model's ability to distinguish between ID and OOD samples by providing a more reliable uncertainty measure compared to traditional Softmax-based methods. Removing this component significantly reduces the model's performance, especially in OOD detection. For example, in Task 2, removing Energy-based scoring drops the AUROC from 84.94% to 71.82%, indicating a reduced ability to separate OOD samples from ID samples.

Additionally, without Energy-based scoring, the FDR increases in all tasks, showing that the model becomes more prone to misclassifying ID samples as OOD. This demonstrates that Energy-based scoring is crucial for reducing false positives and improving the overall reliability of the model.

The ReAct mechanism controls overconfidence by truncating high activations in the penultimate layer, which prevents the model from making highly confident but incorrect predictions on OOD samples. As seen in Table 8, removing ReAct leads to a decrease in AUROC and AUPR, particularly in Task 2, where AUROC drops from 84.94% to 54.30%. This indicates that ReAct plays a critical role in reducing misclassifications of OOD samples.

The combination of Energy-based scoring and ReAct produces a synergistic effect, as shown by the superior performance when both components are used together. For instance, in Task 2, AUROC increases to 84.94% when both Energy and ReAct are applied, compared to 54.30% (Energy only) and 71.82% (ReAct only). This demonstrates that the two components complement each other: Energy-based scoring improves OOD detection, while ReAct reduces overconfidence, ensuring more reliable predictions.

## 3.3 Trustworthy in-distribution classification via OOD detection

In addition to OOD detection, we evaluated the model's performance on a test set containing both ID and OOD samples from the MIT-BIH and INCART datasets. Ensuring trustworthy diagnosis in clinical settings requires not only accurate classification of known diseases but also the ability to identify and handle unknown conditions.

Unlike the Softmax method, which directly classifies all samples in the test set—including OOD samples—often leading to unreliable results, our approach first performs OOD detection to filter out unknown samples. This critical step ensures that only ID samples are passed to the classification stage, significantly improving both the accuracy and reliability of the diagnostic outcomes. By first screening out OOD samples, our method minimizes the risk of misclassification and enhances the trustworthiness of the diagnosis. The following key metrics were used to assess classification performance:

**Accuracy (ACC):** The proportion of correctly classified samples out of the total number of samples.

**Precision:** The proportion of true positive predictions out of all samples predicted as positive, calculated as $\text{Precision}=\dfrac{TP}{TP+FP}$ .

**Recall (Sensitivity):** The proportion of actual positives correctly identified by the model, calculated as $\text{Recall}=\dfrac{TP}{TP+FN}$ .

**F1-Score:** The harmonic means of Precision and Recall, providing a balanced measure of a model's performance, particularly in cases where there is an uneven distribution of classes, calculated as $\text{F1}=2\times\dfrac{\text{Precision}\times\text{Recall}}{\text{Precision}+\text{Recall}}$ .

The classification results are presented in Table 9 for both the MIT-BIH and INCART datasets. For instance, in Task 1 of the MIT-BIH dataset, our method achieved a higher accuracy (89.62%) compared to Softmax (86.68%), although the F1-Score (84.72%) was lower than Softmax (89.62%). This trade-off could be attributed to the model's conservative classification of borderline cases. By being more conservative, our model likely reduced the number of false positives (FP), which directly affects Precision, but this also resulted in missing some true positives (TP), lowering Recall in certain cases. The increase in Accuracy suggests that our method is better at filtering out OOD samples, but at the cost of slightly lower sensitivity to detecting true positives within the ID samples.

In Task 2, our method significantly outperformed Softmax across all metrics, achieving an accuracy of 97.05% and an F1-Score of 95.74%, compared to Softmax's 88.51% accuracy and 83.48% F1-Score. These results indicate that our method excels in this task, likely due to the combined benefits of improved feature extraction through CNN and better handling of class imbalances via the attention mechanisms. Additionally, the OOD detection process effectively filters out irrelevant samples, allowing the model to focus on more confidently classifiable ID data.

For the INCART dataset, our method also outperformed Softmax, achieving an accuracy of 98.15% and an F1-Score of 97.34%, compared to Softmax's 95.51% accuracy and 94.94% F1-Score. These results demonstrate the robustness of our model across different datasets, even when dealing with ECG signals that are more complex or variable. The significant gains in accuracy and F1-Score underscore the model's ability to generalize well and deliver reliable diagnostic outcomes in diverse clinical settings.

**3.3.3 Summary of results.** In summary, our proposed method substantially improves the detection of heart diseases in ECG signals, providing trustworthy diagnoses. Our approach enhances the model's ability to distinguish between known and previously unseen heart disease patterns, reducing false positives while increasing the accuracy and F1-Score in ECG detection.

By accurately identifying unknown heart diseases and maintaining high precision in classifying known ones, our method delivers a reliable and comprehensive diagnostic solution.

**Table 9. Classification results on MIT-BIH and INCART datasets.**

|  | Task | Method | Accuracy | Precision | Recall | F1-Score |
|---|---|---|---|---|---|---|
| MIT-BIH | Task1 | Softmax | 86.68 | 84.79 | 86.68 | 89.62 |
|  |  | Ours | 89.62 | 80.33 | 89.62 | 84.72 |
|  | Task2 | Softmax | 88.51 | 79.00 | 88.51 | 83.48 |
|  |  | Ours | 97.05 | 94.47 | 97.05 | 95.74 |
| INCART | Task3 | Softmax | 95.51 | 94.81 | 95.51 | 94.94 |
|  |  | Ours | 98.15 | 97.57 | 98.15 | 97.34 |

This confirms that applying the ReAct technique and utilizing energy scores can significantly increase the trustworthiness of heart disease detection systems, making them more robust against unknown heart disease types.

## Conclusion

In traditional methods, such as those using Softmax, heart disease classification and detection are often compromised by the model's overconfidence in unknown or out-of-distribution (OOD) data, leading to unreliable and sometimes dangerous diagnostic outcomes. This undermines the ability to provide a truly trustworthy diagnosis.

To resolve this issue and ensure more reliable ECG-based diagnoses, we introduced an OOD detection framework that integrates CNNs with attention mechanisms, alongside Energy and ReAct techniques. This approach significantly improves the model's ability to distinguish known heart diseases from unknown ones, reducing the likelihood of overconfident predictions on OOD samples. By filtering out OOD data before classification, our method ensures that only in-distribution samples are classified, thereby enhancing the trustworthiness of the diagnostic decisions.

Empirical validation using the MIT-BIH Arrhythmia Database and the INCART 12-lead Arrhythmia Database demonstrates that our method not only achieves high sensitivity and specificity for known cardiac conditions but also exhibits exceptional performance in detecting OOD samples. This substantial improvement in both diagnostic precision and generalization capability contributes to a more reliable and trustworthy diagnosis in real-world scenarios, where both known and unknown heart conditions may appear.

By delivering accurate classification of known diseases while safely identifying and excluding unknown conditions, our method establishes a new standard for trustworthy ECG-based diagnosis. This advancement not only reduces the risk of misdiagnosis but also provides strong technical support for the early detection and treatment of heart diseases, ultimately leading to improved patient outcomes.

In terms of clinical integration, our method can be deployed as a decision-support tool within existing ECG analysis workflows. By flagging OOD cases—such as rare or previously unseen heart conditions—our method can assist clinicians in identifying cases that require further investigation or specialist consultation. This will help reduce the risk of over-reliance on automated models, ensuring that clinicians remain actively involved in reviewing uncertain or unfamiliar conditions. The method's ability to provide reliable identification of OOD cases can contribute to more informed clinical decision-making, thus improving diagnostic accuracy and patient safety.

While our method represents a substantial improvement in ECG signal analysis and trustworthy diagnosis, challenges related to potential domain shifts due to individual physiological variability remain. Future efforts will aim to refine our approach to better accommodate individual differences, ensuring broader applicability and effectiveness in personalized clinical settings. Additionally, future work will include real-world clinical validation to assess how seamlessly the method can be integrated into clinical workflows and its impact on patient diagnosis and treatment outcomes.

## Author contributions

**Conceptualization:** Yuhong Liu, Xin Wu, Jing Ren.

**Formal analysis:** Bowen Yu, Zhibin Zhao.

**Investigation:** Bowen Yu, Yuhong Liu, Xin Wu.

**Methodology:** Bowen Yu, Zhibin Zhao.

**Supervision:** Yuhong Liu, Jing Ren.

**Visualization:** Bowen Yu, Zhibin Zhao.

**Writing – original draft:** Bowen Yu.

**Writing – review & editing:** Zhibin Zhao.

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
