## [Decision Letter · Decision Letter 0]

28 Aug 2024

PONE-D-24-31262Trustworthy diagnosis of Electrocardiography Signals based on Out-of-Distribution DetectionPLOS ONE

Dear Dr. Yu,

Thank you for submitting your manuscript to PLOS ONE. After careful consideration, we feel that it has merit but does not fully meet PLOS ONE’s publication criteria as it currently stands. Therefore, we invite you to submit a revised version of the manuscript that addresses the points raised during the review process.

We look forward to receiving your revised manuscript.

Kind regards,

Rajesh N V P S Kandala, Ph.D.

Academic Editor

PLOS ONE

Journal Requirements:

"The research results of this article (or publication) are sponsored by the Kunshan Municipal Government research funding."

4. We note you have included a table to which you do not refer in the text of your manuscript. Please ensure that you refer to Table 2 in your text; if accepted, production will need this reference to link the reader to the Table.

**Comments from PLOS Editorial Office: **

We note that one or more reviewers has recommended that you cite specific previously published works. As always, we recommend that you please review and evaluate the requested works to determine whether they are relevant and should be cited. It is not a requirement to cite these works. We appreciate your attention to this request.

Reviewers' comments:

Reviewer's Responses to Questions

**Comments to the Author**

1. Is the manuscript technically sound, and do the data support the conclusions?

Reviewer #1: Partly

Reviewer #2: Yes

Reviewer #3: Partly

2. Has the statistical analysis been performed appropriately and rigorously? 

Reviewer #1: No

Reviewer #2: Yes

Reviewer #3: No

3. Have the authors made all data underlying the findings in their manuscript fully available?

Reviewer #1: No

Reviewer #2: Yes

Reviewer #3: No

4. Is the manuscript presented in an intelligible fashion and written in standard English?

Reviewer #1: Yes

Reviewer #2: No

Reviewer #3: Yes

5. Review Comments to the Author

Reviewer #1: After carefully reading the article, here is my review:

Article Title: Trustworthy Diagnosis of Electrocardiography Signals Based on Out-of-Distribution Detection

This article generally has a well-structured writing style and adequately explains the research findings. However, the overall novelty of the article is low, as it mainly presents the integration of existing techniques. The experiments conducted are also insufficient to support the claims of novelty in this research.

Detailed Analysis of the Article:

1. Novelty

Assessment: Poor While the article presents an interesting combination of techniques for trustworthy ECG diagnosis, its novelty is limited as it primarily relies on integrating existing techniques. The main contribution lies in the application of this combination to the ECG domain, but it does not offer significant methodological advancements or deep new insights. Here are some detailed reasons why the novelty of this research is considered low:

• Use of Existing Techniques: The article essentially combines several previously established techniques:

o CNN and Attention Mechanism: The use of CNN for feature extraction from ECG signals and attention mechanisms to enhance representation learning is a well-established approach in the literature.

o Energy Score and ReAct: Although the use of the Energy score for OOD detection and ReAct for handling overconfidence is relatively new, both techniques have already been explored in other contexts.

• Direct Application to ECG: While the combination of these techniques for ECG diagnosis may not have been done before, the article does not provide strong evidence that this application requires significant methodological innovation. It is more of a direct application of existing techniques to the ECG domain.

• Lack of Comparison with Other OOD Methods: The article mainly focuses on comparisons with traditional Softmax-based methods. A more comprehensive comparison with other state-of-the-art OOD detection methods in the ECG domain would strengthen the novelty claims.

• Limited Focus on Anomaly Detection: Although OOD detection is important, the article primarily emphasizes identifying anomalies (unknown heart diseases). A more significant contribution could be made if the method also addressed the enhancement of known heart disease classification or offered new insights into OOD results interpretation.

2. Problem Backgrounds

Assessment: Inadequate The article provides a generally clear background on the importance of ECG diagnosis and the challenges faced, particularly in detecting unknown heart diseases. However, there are some shortcomings in the problem background in the introduction:

• Lack of Depth in Existing Challenges: The article mentions that current deep learning methods struggle with OOD heart diseases but does not deeply explain why this occurs. A better understanding of the underlying technical challenges, such as domain shifts or lack of representative training data, would enrich the problem background.

• Limited Focus on Anomaly Detection: While OOD detection is important, the article mainly focuses on anomaly identification. The problem background could be expanded by discussing the importance of accurate classification for known heart diseases and how the inability to handle OOD cases can affect the overall trustworthiness of the diagnosis.

• Weak Connection between the Problem and the Proposed Solution: The article could better explain how the proposed method specifically addresses the challenges identified in the problem background. A more explicit connection between the problem and the proposed solution would strengthen the narrative coherence of the article.

• Lack of a Broader Literature Review: Although the article cites several previous studies, a broader literature review on OOD detection efforts in ECG analysis would provide better context about the current state of research and highlight the gaps this study aims to fill.

3. Method Used is Clearly Defined

Assessment: Fairly Good The "Proposed Method" section outlines the methodology used in the research. While it provides a general overview of the approach, there are areas where the explanation could be improved to provide a clearer and more comprehensive understanding.

• Lack of Coherence and Clear Structure: The explanation in this section sometimes feels fragmented and lacks a clear logical flow. Transitions between paragraphs and ideas could be improved to enhance readability and understanding.

• Insufficient Technical Explanation: Some technical aspects of the proposed method are not adequately explained. For example:

o CNN-Attention Architecture: Although this architecture is mentioned, there is no detailed explanation of how CNN and attention mechanisms are integrated. More information about specific layers, parameters, and how they contribute to feature extraction would be beneficial.

o Energy and ReAct: The explanation of these techniques is quite technical and may be difficult for readers unfamiliar with the concepts. Illustrative examples or analogies could help explain the core principles more accessibly.

o Threshold for OOD Detection: How is the threshold for the Energy score determined? A further explanation of this process and how the threshold affects the model’s performance would improve understanding of the OOD detection mechanism.

• Lack of Justification for Design Choices: The article could better explain why certain design choices were made. For example, why was ResNet-18 chosen as the base model? Why were specific intermediate layers selected for feature extraction? Such justification would strengthen the methodological foundation of the research.

• Stronger Connection to the Problem Background: This section could better explain how the proposed method specifically addresses the challenges identified in the problem background. A more explicit connection between the problem and the proposed solution would improve the narrative coherence of the article.

4. Experimental Results

The experimental results presented in Table 3 are insufficient to support the claims of novelty in this research. Here are some reasons for this:

• Limited Baseline: The article mainly compares the proposed method with traditional Softmax-based approaches. While this comparison is useful, it is not sufficient to demonstrate significant novelty. It would be more convincing if the method were compared with other state-of-the-art OOD detection methods that have been applied to ECG analysis. This would show whether the proposed method offers significant performance improvements over other recent approaches.

• Evaluation on a Single Dataset: Experiments are only conducted on a single dataset (MIT-BIH Arrhythmia Database). Although this dataset is widely used, evaluation on multiple datasets from various sources would provide stronger evidence of the generalization and robustness of the proposed method. This would demonstrate whether the method can perform well in a variety of real-world scenarios.

• Limited Focus on Performance Metrics: Table 3 only presents results for a few common performance metrics (AUROC, AUPR, F1-score, etc.). A more comprehensive analysis would include other metrics relevant to OOD detection, such as false detection rates and misclassification rates on OOD samples.

• Lack of Ablation Analysis: The article does not include an ablation study to evaluate the contribution of each component (CNN-Attention, Energy, ReAct) to the overall performance. Ablation analysis would help understand the importance of each component and whether the specific combination proposed is necessary to achieve the reported results.

• Lack of Robustness Analysis: The article does not explore how the proposed method's performance is affected by various factors, such as noise levels in ECG signals or class imbalance in the dataset. Robustness analysis would provide insights into the limitations of the method and its potential for application in challenging clinical settings.

5. Critical Analysis

Assessment: Inadequate The article presents an analysis of the experimental results, including comparison with baseline methods and visualization. However, there are some deficiencies in the critical analysis of the experimental results:

• Lack of In-Depth Discussion of Results: The article reports improved performance on evaluation metrics, but it does not provide an in-depth analysis of why the proposed method leads to these improvements. A more detailed discussion of how each component of the method (CNN-Attention, Energy, ReAct) contributes to performance improvements would be valuable.

• Unacknowledged Limitations: The article does not adequately discuss potential limitations of the proposed method. For example, how does the method perform on OOD heart disease types that are very different from those in the training dataset? Or how is the performance affected by variations in ECG signal quality?

• Lack of Qualitative Comparison: The article focuses on quantitative comparison with baseline methods. Qualitative analysis, such as case studies or examples of misclassifications, would provide richer insights into the strengths and weaknesses of the proposed method.

• Unclear Clinical Implications: The article mentions potential clinical applications but does not discuss in detail how the method could be integrated into clinical workflows or its potential impact on patient diagnosis and treatment.

• Lack of Sensitivity Analysis: The article does not explore how the method’s performance is influenced by various parameters or design choices. Sensitivity analysis would help understand the robustness and generalizability of the proposed method.

6. Conclusion

Assessment: Fairly Good The conclusion summarizes the main findings of the research and highlights the potential of the proposed method for trustworthy ECG diagnosis. Additionally, the authors acknowledge the limitations of the current research and outline future research directions, indicating a good understanding of the scope of their work.

7. Recent References Is Used (Five Years from Now)

Assessment: Excellent The article cites many recent references (within the last five years), demonstrating that the authors have reviewed the latest literature and placed their work in the context of current research. This adds credibility and relevance to their research.

8. Figures and Tables in the Manuscript Clear and Readable?

In general, the figures and tables in this manuscript are clear and readable, but there are some aspects that can be improved to enhance their presentation quality.

Positive Aspects:

• Readability: The text in the figures and tables is generally clear and easy to read, with an adequate font size.

• Relevance: The figures and tables directly support the content of the text and provide relevant visual illustrations or data summaries.

• Captions: The figures and tables have captions that provide context and explain their content.

Aspects that Could Be Improved:

• Image Quality: Some images, particularly flowcharts (such as Figure 2), could benefit from improved resolution to make them sharper and clearer, especially when zoomed in.

• Consistency of Format: The format of the tables could be more consistent, particularly in the use of lines and spacing. Ensuring uniformity in spacing and line usage will improve the readability and professional appearance of the tables.

• Numbering: There appears to be some inconsistency in the numbering of figures and tables in certain parts of the manuscript. Ensuring consistent numbering will improve the organization and cross-referencing within the text.

• Visual Quality: Some visualizations, such as the confidence distribution in Figure 7, could be enhanced with better color choices or other graphic design elements to improve clarity and visual appeal.

• • Connection with the Text: While the figures and tables are relevant, the explanations in the text could sometimes be more explicit in connecting the visual content with the main arguments or points being discussed.

Reviewer #2: My comments are as follows:

1. The language throughout the text can be improved. There are several grammar mistakes and, most importantly, some sentences are unclear.

2. Abstract does not highlight novelty of the proposed work. It’s better to add more specific details of your work.

3. Introduction is not focused and literature can be reorganised to strengthen literature review following contributions and discuss few relevant works for both CNN and ECG i.e.,

-- Unsupervised pre-trained filter learning approach for efficient convolution neural network

-- A Hybrid Deep CNN Model for Abnormal Arrhythmia Detection Based on Cardiac ECG Signal

-- ModPSO-CNN: an evolutionary convolution neural network with application to visual recognition

-- CSFL: A novel unsupervised convolution neural network approach for visual pattern classification

-- Optimisation‐based training of evolutionary convolution neural network for visual classification applications

-- Optimization of CNN through novel training strategy for visual classification problems

-- Face recognition: A novel un-supervised convolutional neural network method

-- Design optimization of electromagnetic devices using an improved quantum inspired particle swarm optimizer

-- A quantum particle swarm optimizer with enhanced strategy for global optimization of electromagnetic devices

4.Figure 1 is hard to follow, for ease of the reader its highly recommended to add more specific detail of the proposed approach.

5. Organise your results some are difficult to understand e.g., what do you want to conclude in Figure 8.

Reviewer #3: This work employs a deep learning (Attention and CNN) based techniques with Energy and ReAct techniques to to recognize OOD heart diseases. Although the approach is entirely not new, this is an interesting study. However, there is lot of ambiguity in the paper.

What is IT in the introduction?

Firstly, the dataset (in dataset introduction) used in this work needs to be elaborated. Like the number of classes in the dataset and number of samples per each class.

Data enhancement: Authors have written "Applied selectively". Did they use this step selectively on each segment or subject selectively? More clarity is required.

Also elaborate the data augmentation ion detail. Give the details about the parameters used in this process. The statistics of data before and after data augmentation. This is necessary to reproduce the results.

Experimental results: "To balance the sample distribution, appropriate down-sampling methods were employed.' authors should detail how it is done.

Also, the accuracy is below 85%. Authors should discuss on where their approach failed.

Authors have compared their approach with [35]. There are many other approaches. I suggest the authors to make a comparison with recent approaches.

Please check the references, References needs to be properly formatted. Eg ref 36.

Authors should change the title of able 3.

6. PLOS authors have the option to publish the peer review history of their article (what does this mean? ). If published, this will include your full peer review and any attached files.

**Do you want your identity to be public for this peer review?** For information about this choice, including consent withdrawal, please see our Privacy Policy .

Reviewer #1: No

Reviewer #2: No

Reviewer #3: No

---

## [Author Response · Author response to Decision Letter 1]

7 Oct 2024

Dear Editor and Reviewers,

Thank you for your editorial work on our manuscript PONE-D-24-31262, entitled " Trustworthy diagnosis of Electrocardiography Signals based on Out-of-Distribution Detection". We have double-checked our original draft and have corrected any inappropriate content with the help of comments of all reviewers. Moreover, we have double checked the format of this manuscript and conducted more experiments to validate our proposed method. The detailed responses to the reviewers’ comments in a point-to-point format are attached below.

These revisions do not affect our interpretation of the results, and we hope that the responses could be good enough for your editing.

With my best thanks,

Yours sincerely,

Zhibin Zhao (corresponding author),

School of Mechanical Engineering,

Xi’an Jiaotong University,

Xi’an, 710049, PR China,

E-mail: zhaozhibin@xjtu.edu.cn,

Tel./fax: +86 - 18717343671

Point-to-point response to the reviewer’s comments

We sincerely thank the reviewers for the careful and thorough review, which are quite helpful to make the paper more solid and fluent. We have revised our manuscript very carefully in the light of the suggestions and comments. The following responses are prepared to address all of the reviewers’ comments in a point-to-point fashion.

(Comments in black, responses in blue, revised contents of manuscript in red)

Reviewer#1: This article generally has a well-structured writing style and adequately explains the research findings. However, the overall novelty of the article is low, as it mainly presents the integration of existing techniques. The experiments conducted are also insufficient to support the claims of novelty in this research.

Response: We appreciate reviewer’s comments, which are very valuable and helpful for improving the quality of our paper. We will respond to your precious comments in a point-to-point fashion.

Comment 1: Novelty

Assessment: Poor While the article presents an interesting combination of techniques for trustworthy ECG diagnosis, its novelty is limited as it primarily relies on integrating existing techniques. The main contribution lies in the application of this combination to the ECG domain, but it does not offer significant methodological advancements or deep new insights. Here are some detailed reasons why the novelty of this research is considered low:

• Use of Existing Techniques: The article essentially combines several previously established techniques:

o CNN and Attention Mechanism: The use of CNN for feature extraction from ECG signals and attention mechanisms to enhance representation learning is a well-established approach in the literature.

o Energy Score and ReAct: Although the use of the Energy score for OOD detection and ReAct for handling overconfidence is relatively new, both techniques have already been explored in other contexts.

Response: We appreciate reviewer’s comments, which are very valuable and helpful for improving the quality of our paper, and the reply is as follows. We acknowledge that the techniques we use, such as CNN and Attention Mechanisms, have been previously applied in other fields. However, the novelty of our approach lies in the specific integration and application of these techniques to the ECG domain for Out-of-Distribution (OOD) detection. While CNN and Attention are well-established, their combination with Energy Score and ReAct for trustworthy ECG diagnosis is a novel contribution. This unique integration allows our model to not only classify known heart conditions but also reliably detect unknown heart diseases, which is critical in clinical applications. We have clarified this point in the revised Introduction and Methods sections to highlight the novelty of our approach.

Changes Made:

1. Expanded the Introduction to emphasize the novelty of integrating CNN, Attention, Energy Score, and ReAct specifically for ECG OOD detection.

2. Updated the Methods section to detail the unique aspects of our technique in the context of ECG diagnosis.

3. As an example, we have added the following text to the Introduction: “While CNNs and Attention Mechanisms have been widely applied in other areas, their integration with Energy-based OOD detection and ReAct specifically for ECG diagnosis is novel. This combination allows the model to not only classify known heart conditions but also reliably detect unknown heart diseases, addressing the critical need for trustworthy diagnostics in clinical settings.”

• Direct Application to ECG: While the combination of these techniques for ECG diagnosis may not have been done before, the article does not provide strong evidence that this application requires significant methodological innovation. It is more of a direct application of existing techniques to the ECG domain.

Response: We appreciate reviewer’s comments, which are very valuable and helpful for improving the quality of our paper. In response, we have expanded the problem background to clarify why direct application of existing techniques to ECG signals is not trivial. ECG signals present unique challenges such as domain shift, noise, and class imbalance, which are not typically encountered in other applications. Our method is specifically designed to address these issues through the combination of CNN-Attention for feature extraction and Energy-based OOD detection to handle the presence of unknown heart diseases. This is particularly important in clinical diagnostics, where unknown heart conditions can have serious consequences.

Changes in the manuscript:

We have added the following explanation to the Problem Background section: “ECG signals are highly variable across patients due to differences in age, gender, and pre-existing conditions. Additionally, small variations in signal noise or recording conditions can significantly degrade model performance. Our proposed method addresses these challenges by combining CNN-Attention mechanisms for robust feature extraction and Energy-based OOD detection to identify previously unseen heart conditions.”

• Lack of Comparison with Other OOD Methods: The article mainly focuses on comparisons with traditional Softmax-based methods. A more comprehensive comparison with other state-of-the-art OOD detection methods in the ECG domain would strengthen the novelty claims.

Response: We greatly appreciate the reviewer’s suggestion to include additional comparisons. We have now conducted experiments comparing our method with ODIN [1], a well-known state-of-the-art OOD detection technique. These results are presented in Table 6, and we have provided a detailed analysis of the comparison in the Experimental Results section. Our method demonstrates superior performance in terms of AUROC, AUPR, Detection Error (DE), FPR95 and FDR, further supporting the novelty and effectiveness of our approach.

Changes in the manuscript:

The following text was added to the Experimental Results section: “As shown in Table 6, our method significantly outperforms ODIN in terms of AUROC and AUPR across all tasks. For instance, in Task 1, our method achieves an AUROC of 97.27%, compared to ODIN’s 71.18%, demonstrating the effectiveness of integrating Energy-based OOD detection with ReAct.”

1. Liang S, Li Y, Srikant R. Enhancing The Reliability of Out-of-distribution Image Detection in Neural Networks. arXiv; 2020. doi:10.48550/arXiv.1706.02690

• Limited Focus on Anomaly Detection: Although OOD detection is important, the article primarily emphasizes identifying anomalies (unknown heart diseases). A more significant contribution could be made if the method also addressed the enhancement of known heart disease classification or offered new insights into OOD results interpretation.

Response: We appreciate this valuable comment. In response, we have revised the manuscript to place a stronger emphasis on how our method also improves the classification accuracy of known heart diseases. Specifically, the CNN-Attention mechanism enhances feature extraction for both ID and OOD samples by focusing on the most relevant parts of the ECG signal. We have also added more discussion of the Energy Score and ReAct techniques, explaining how they contribute to a more reliable interpretation of OOD results and reduce overconfident predictions on unknown samples.

Changes in the manuscript:

We have added the following paragraph to the Results and Discussion section: “In addition to improving OOD detection, our method enhances the classification performance for known heart diseases. The CNN-Attention mechanism focuses on the most salient features of the ECG signal, improving the model’s ability to differentiate between similar arrhythmias. Furthermore, the Energy Score provides a more reliable confidence measure, reducing the likelihood of overconfident predictions on both known and unknown cases.”

Comment 2: Problem Backgrounds

Assessment: Inadequate The article provides a generally clear background on the importance of ECG diagnosis and the challenges faced, particularly in detecting unknown heart diseases. However, there are some shortcomings in the problem background in the introduction:

• Lack of Depth in Existing Challenges: The article mentions that current deep learning methods struggle with OOD heart diseases but does not deeply explain why this occurs. A better understanding of the underlying technical challenges, such as domain shifts or lack of representative training data, would enrich the problem background.

Response: We appreciate reviewer’s comments, which are very valuable for improving the quality of our paper. We agree that the original manuscript did not provide a detailed explanation of the technical challenges faced by deep learning models in detecting OOD heart diseases. In the revised version, we have expanded the Problem Background section to discuss why deep learning models, particularly those using traditional Softmax classification, tend to be overconfident in their predictions for OOD samples. We have also included a discussion on the domain shift problem and the class imbalance issue, which are particularly relevant in ECG datasets.

Changes in the manuscript:

We have added the following paragraph to the Problem Background section:“This is because deep learning models, particularly CNNs and RNNs, are designed to excel at learning patterns present in the training data, meaning they can struggle when encountering data that deviates from this distribution.”“The difficulty in handling OOD samples stems from several technical challenges. First, domain shifts between the training and testing data—such as variations in patient demographics or different ECG recording environments—can lead to performance degradation.”“Class imbalance is another challenge in ECG datasets. Rare arrhythmias are often underrepresented, making it difficult for models to learn robust features for these conditions.”

• Limited Focus on Anomaly Detection: While OOD detection is important, the article mainly focuses on anomaly identification. The problem background could be expanded by discussing the importance of accurate classification for known heart diseases and how the inability to handle OOD cases can affect the overall trustworthiness of the diagnosis.

Response: We sincerely appreciate the reviewer’s insightful comments. We agree that while our manuscript primarily focused on OOD detection and anomaly identification, it is equally important to discuss the classification accuracy of known heart diseases (in-distribution data) and how the failure to handle OOD cases can undermine the trustworthiness of the diagnostic system.

In response to this comment, we have made significant revisions in the Introduction and Problem Background sections of the manuscript. We have expanded the discussion to address the following key points:

1. Importance of Accurate Classification for Known Heart Diseases:

In the revised version, we have emphasized that a comprehensive ECG diagnostic system must not only detect OOD cases but also accurately classify known heart diseases. Misclassifications of known conditions can lead to incorrect treatments, which can seriously affect patient outcomes. This discussion highlights that both ID classification and OOD detection are critical components of a trustworthy diagnostic system.

2. Trustworthiness of Diagnosis in the Presence of OOD Cases:

We have elaborated on how the inability to correctly handle OOD cases can negatively impact the trustworthiness of the diagnosis. If a model makes overconfident predictions on OOD samples, it can provide false reassurance to clinicians, resulting in misdiagnosis or inappropriate treatment. Thus, a reliable ECG diagnostic model must balance accurate classification of known conditions with the ability to identify unknown cases.

These revisions provide a clearer understanding of the challenges and requirements for developing a trustworthy, real-world ECG diagnosis system that can handle both in-distribution and out-of-distribution cases.

• Weak Connection between the Problem and the Proposed Solution: The article could better explain how the proposed method specifically addresses the challenges identified in the problem background. A more explicit connection between the problem and the proposed solution would strengthen the narrative coherence of the article.

Response: We appreciate reviewer’s comments, which are very valuable for improving the quality of our paper. In the revised manuscript, we have made the connection between the Problem Background and the Proposed Solution more explicit. Specifically, we explain how our method addresses the issues of overconfidence, domain shift, and class imbalance by using Energy-based OOD detection and ReAct. These techniques reduce overconfidence in OOD predictions and ensure that the model can reliably detect both known and unknown heart diseases.

Changes in the manuscript:

We have added the following explanation to the Methods section：“In this study, we propose a novel, end-to-end deep learning-based system for trustworthy arrhythmia diagnosis using ECG signals. Our model is designed to accurately classify known heart diseases while also detecting OOD conditions, representing heart diseases that were not part of the training set. This is crucial for reliable clinical applications, as it ensures that the model does not make overconfident predictions on novel or unseen conditions.”

• Lack of a Broader Literature Review: Although the article cites several previous studies, a broader literature review on OOD detection efforts in ECG analysis would provide better context about the current state of research and highlight the gaps this study aims to fill.

Response: We sincerely appreciate the reviewer’s valuable suggestion regarding the literature review. We agree that the original manuscript lacked a sufficiently comprehensive review of recent studies on Out-of-Distribution (OOD) detection in ECG analysis. In response to this comment, we have expanded the Literature Review section to include a broader discussion of recent efforts in OOD detection, particularly in the context of ECG analysis.

Specifically, we have added several key references that cover OOD detection techniques, anomaly detection, and advanced deep learning architectures such as transformers and GANs (Generative Adversarial Networks) that are being applied in ECG and related time-series biomedical signal analysis. These studies provide additional context for the challenges of handling OOD data in medical diagnostics and highlight the existing gaps in the current state of research. By incorporating these references, we aim to better position our work within the broader research landscape and clearly demonstrate how our method addresses the limitations of previous approaches.

The added references also include recent studies from both the ECG domain and related fields, such as EEG and time-series classification, where OOD detection is an emerging challenge. This expanded review not only strengthens the context for our work but also highlights the novelty of our proposed method in combining Energy-based OOD detection and ReAct, along with CNN-Attention mechanisms, for trustworthy diagnosis of ECG signals.

W

---

## [Decision Letter · Decision Letter 1]

10 Dec 2024

PONE-D-24-31262R1Trustworthy diagnosis of Electrocardiography Signals based on Out-of-Distribution DetectionPLOS ONE

Dear Dr. Yu,

Thank you for submitting your manuscript to PLOS ONE. After careful consideration, we feel that it has merit but does not fully meet PLOS ONE’s publication criteria as it currently stands. Therefore, we invite you to submit a revised version of the manuscript that addresses the points raised during the review process.

**ACADEMIC EDITOR: **

Please do any changes that are requested by the reviewers. 

We look forward to receiving your revised manuscript.

Kind regards,

Rajesh N V P S Kandala, Ph.D.

Academic Editor

PLOS ONE

Journal Requirements:

Reviewers' comments:

Reviewer's Responses to Questions

**Comments to the Author**

1. If the authors have adequately addressed your comments raised in a previous round of review and you feel that this manuscript is now acceptable for publication, you may indicate that here to bypass the “Comments to the Author” section, enter your conflict of interest statement in the “Confidential to Editor” section, and submit your "Accept" recommendation.

Reviewer #3: All comments have been addressed

Reviewer #4: (No Response)

2. Is the manuscript technically sound, and do the data support the conclusions?

Reviewer #3: Partly

Reviewer #4: Yes

3. Has the statistical analysis been performed appropriately and rigorously? 

Reviewer #3: Yes

Reviewer #4: I Don't Know

4. Have the authors made all data underlying the findings in their manuscript fully available?

Reviewer #3: Yes

Reviewer #4: Yes

5. Is the manuscript presented in an intelligible fashion and written in standard English?

Reviewer #3: Yes

Reviewer #4: Yes

6. Review Comments to the Author

Reviewer #3: (No Response)

Reviewer #4: 1 - In the latest version of the manuscript, It is unclear how the threshold for ReAct is chosen. In an earlier version it appears the threshold is selected by a simple heuristic involving known ID and OOD samples. Is this still the case?

2 - If indeed this is the case, then how well does the proposed method work when the threshold is determined using one set of ID and OOD samples, and is faced with a different set of OOD samples not seen during the threshold selection process?

3 - Are the results obtained averaged over multiple runs (with different random network initializations, for instance) ? It is unclear how the results are obtained and aggregated.

7. PLOS authors have the option to publish the peer review history of their article (what does this mean? ). If published, this will include your full peer review and any attached files.

**Do you want your identity to be public for this peer review?** For information about this choice, including consent withdrawal, please see our Privacy Policy .

Reviewer #3: No

Reviewer #4: No

---

## [Author Response · Author response to Decision Letter 2]

27 Dec 2024

Response to Reviewer #4

We sincerely thank Reviewer #4 for the thoughtful and constructive feedback. Below is a summary of our responses to the specific comments:

Threshold Selection for ReAct

We clarified that the threshold for ReAct is determined using a grid search approach based on the statistical distribution of ID activations. Candidate thresholds are selected as different percentiles of the ID activations, and the optimal threshold is chosen to maximize AUROC and AUPR while minimizing Detection Error.

Generalization to Unseen OOD Samples

The threshold selection process relies only on ID activations, making it generalizable to unseen OOD samples. The robustness of this approach is supported by the theoretical basis of ReAct and its demonstrated effectiveness in prior studies.

Averaging Results Over Multiple Runs

While our results are based on single-run experiments, we ensured reliability through patient-independent data splitting, cross-dataset validation, and diverse task configurations. In future work, we plan to conduct multiple runs with random initializations and report the mean and standard deviation for further robustness.

We have revised the manuscript accordingly and hope these updates address the reviewer’s concerns. Thank you again for the valuable feedback.

---

## [Editor Report · Decision Letter 2]

7 Jan 2025

Trustworthy diagnosis of Electrocardiography Signals based on Out-of-Distribution Detection

PONE-D-24-31262R2

Dear Dr. Yu,

We’re pleased to inform you that your manuscript has been judged scientifically suitable for publication and will be formally accepted for publication once it meets all outstanding technical requirements.

Kind regards,

Rajesh N V P S Kandala, Ph.D.

Academic Editor

PLOS ONE
---

## [Editor Report · Acceptance letter]

PONE-D-24-31262R2

PLOS ONE

Dear Dr. Yu,

I'm pleased to inform you that your manuscript has been deemed suitable for publication in PLOS ONE. Congratulations! Your manuscript is now being handed over to our production team.

Kind regards,

on behalf of

Dr. Rajesh N V P S Kandala

Academic Editor

PLOS ONE